# Assessment of Gold Bio-Functionalization for Wide-Interface Biosensing Platforms

**DOI:** 10.3390/s20133678

**Published:** 2020-06-30

**Authors:** Lucia Sarcina, Luisa Torsi, Rosaria Anna Picca, Kyriaki Manoli, Eleonora Macchia

**Affiliations:** 1Dipartimento di Chimica, Università degli Studi di Bari Aldo Moro, 70125 Bari, Italy; lucia.sarcina@uniba.it (L.S.); luisa.torsi@uniba.it (L.T.); rosaria.picca@uniba.it (R.A.P.); 2CSGI (Centre for Colloid and Surface Science), Department of Chemistry, 70125 Bari, Italy; 3The Faculty of Science and Engineering, Åbo Akademi University, FI-20500 Turku, Finland; eleonora.macchia@abo.fi

**Keywords:** surface plasmon resonance, biosensors, bio-functionalization optimization, cost-effective biosensors, lab-on-a-chip

## Abstract

The continuous improvement of the technical potential of bioelectronic devices for biosensing applications will provide clinicians with a reliable tool for biomarker quantification down to the single molecule. Eventually, physicians will be able to identify the very moment at which the illness state begins, with a terrific impact on the quality of life along with a reduction of health care expenses. However, in clinical practice, to gather enough information to formulate a diagnosis, multiple biomarkers are normally quantified from the same biological sample simultaneously. Therefore, it is critically important to translate lab-based bioelectronic devices based on electrolyte gated thin-film transistor technology into a cost-effective portable multiplexing array prototype. In this perspective, the assessment of cost-effective manufacturability represents a crucial step, with specific regard to the optimization of the bio-functionalization protocol of the transistor gate module. Hence, we have assessed, using surface plasmon resonance technique, a sustainable and reliable cost-effective process to successfully bio-functionalize a gold surface, suitable as gate electrode for wide-field bioelectronic sensors. The bio-functionalization process herein investigated allows to reduce the biorecognition element concentration to one-tenth, drastically impacting the manufacturing costs while retaining high analytical performance.

## 1. Introduction

Single-molecule detection is a crucial task to accomplish [1,2], which could in fact allows to gather digital tracking of a biomarker from its physiologic to its pathogenic level. Such an analytical tool will enable to define the onset from a healthy to diseased patient. Early diagnostics in progressive diseases would, hence, become possible well before any symptom appears. Single-molecule biomarker detection would further allow marker quantification non-invasively, in readily available biofluids such as saliva, sweat, or even tears where they can be present at much lower concentrations. Along the same line, it would make possible ultrasensitive liquid biopsy, i.e., the assay of peripheral biofluids such as plasma, serum, or even saliva, a feasible medical procedure replacing the invasive inspection of diseased tissues. Among the single-molecule detection methods proposed so far, only a few are exploitable for real clinical sensing. This approach, addressed as wide-field sensing [2], involves the assay of a biomarker at the attomolar (aM, 10^−18^ M) or even zeptomolar (zM, 10^−21^ M) limit-of-detection with a large-area interface that is functionalized with a huge number of biorecognition elements (10^11^–10^12^ cm^−2^). Large-area organic-bioelectronic devices endowed with an electronic-interface capable to detect recognition element/biomarker complexes intrinsic properties, such as the electrostatic or dielectric ones, are emerging as a powerful tool capable of selective, label-free, and fast biomarker detection at the physical limit in real biofluids. Bioelectronic thin-film transistors (TFTs) [3,4,5], gated via an ionically-conducting and electronically-insulating electrolyte [6,7,8], are generating a lot of interest as they can potentially be produced by scalable large-area low-cost approaches. Such sensors [9,10] are capable of high selectivity via the bio-functionalization of the organic semiconductors (OSCs) [10] or the gate metal surface [11]. Electrolyte-gated TFTs (EG-TFTs) have been successfully engaged, lately, as wide-field bioelectronic sensors exhibiting limits of detection at zM - aM level also in real bio-fluids [12,13]. In the Single-Molecule with a large-Transistor (SiMoT) platform, based on an EG-TFT device [12], the gate is bio-functionalized [14] with 10^12^ cm^−2^ recognition elements covalently attached to a 0.5 cm^2^ gold gate. The SiMoT platform has been demonstrated to successfully perform label-free detection at the physical limit of immunoglobulin-M [13], C-reactive protein in saliva [15], and HIV-p24 [16,17], as well as genomic biomarkers [18]. However, in clinical practice, to gather enough information to formulate a diagnosis, multiple biomarkers are normally quantified from the same biological sample. Therefore, it is necessary to develop a technology capable to perform multiplexing [19] and thus realized with an array of 96 or more transducing elements, so that the standard solutions, the negative-controls, and the sample can be assayed, with all the replicates for each biomarker, at the same time. All this calls for translating the lab-based SiMoT device into a cost-effective portable multiplexing array prototype that integrates, with a modular approach, novel materials, and standard components/interfaces. In this perspective, the assessment of cost-effective manufacturability represents a crucial step, with specific regard to the optimization of the bio-functionalization protocol of the gate modules. Therefore, in this study, we have developed, with the aid of surface plasmon resonance (SPR) technique, a novel and reliable cost-effective process to successfully bio-functionalize a gold surface suitable as gate electrode in a SiMoT-based platform. An uncommon approach has been chosen in the SPR settings by using a method that can be easily translated to bioelectronic device surface modifications. In particular, the bio-functionalization process herein proposed allows to reduce to one-tenth the concentration of the biorecognition elements. This can drastically reduce the manufacturing costs without sacrificing the sensing performance in terms of sensitivity. In general, for a biomolecular reaction to occur, the two reagents need to be confined in an adequately small volume for a sufficiently long time. The recognition element can be attached on a surface that serves as detecting interface. Regardless, the interaction cross-section of the two reagents has to be reasonably high. In this respect, a volume of 1 μm^3^ (1 femtoliter - fL) has been proven sufficiently small for a single enzyme to interact with its substrate (present in excess concentration though) on the minute time-scale [20,21]. Indeed, a solution comprising n = 1 ± 1 (√n = Poisson error) molecules in each 1fL sub-volumes has a concentration of ~ 1 × 10^−9^ mol × l^−1^ (nM). Smaller volumes (attoliter, aL, or zeptoliter, zL), each occupied by a single molecule, entails even larger concentrations. Since the number of molecules in a volume V = 100 μL of a solution of molar concentration [c] is n = [c]·V·N_A_ (N_A_ = Avogadro’s number), 1 nM equals ~10^11^ molecules or, equivalently, ~10^11^ 1 fL sub-volumes. As two molecules need to be confined in a volume of 1 fL or smaller to rapidly interact, at least one of them is to be present at a concentration of 1 nM. Every 1 fL (or lower) statistically contains one reagent, so wherever the other single reagent is, there is always one fL sub-volume comprising both reagents. A single-molecule interaction can, therefore, occur when the recognition-elements are present at nM concentration (or higher) along with a single biomarker or the opposite way around. In clinical assays, the former is to be preferred. Therefore, the aim of the study herein presented is to determine which is the minimum biorecognition element concentration requested to achieve a sufficiently high surface coverage. Interestingly, it has been demonstrated that it is possible to reduce to one tenth the biorecognition elements concentration without impacting on the analytical performances, thus optimizing the biofunctionalization protocol of the SiMBiT platform as well as of other bioelectronic devices [22,23,24]. This study paves the way toward a multiplexing single molecule technology that will open to a massive use of high-throughput array-based assay not only in clinical laboratory analysis but also in point-of-care and low resources settings.

## 2. Materials and Methods

### 2.1. Materials

3-mercaptopropionic acid (3-MPA), 11-mercaptoundecanoic acid (11-MUA), 1-ethyl-3-(3-dimethylaminopropyl)-carbodiimide (EDC), N-hydroxysulfosuccinimide sodium salt (NHSS), ethanolamine hydrochloride (EA), and 66 kDa molecular weight bovine serum albumin (BSA) were purchased from Sigma–Aldrich and used with no further purification. HPLC-grade water, ethanol grade puriss. p.a. assay, ≥99.8%, ammonium hydroxide solution (NH_4_OH) 28.0–30.0%, hydrogen peroxide (H_2_O_2_) 30% (w/w) in H_2_O were purchased from Sigma–Aldrich and used with no further purification. Anti-human immunoglobulin M (anti-IgM) produced in goat polyclonal antibodies was purchased from Sigma–Aldrich and used with no further purification. Human IgM (∼950 kDa) affinity ligand, purchased from Sigma–Aldrich, was isolated from pooled normal human serum and used with no further purification. Phosphate buffered saline (PBS pH 7.4, Sigma-Aldrich- Merck KGaA, Darmstadt, Germany) solution was prepared according to previous works [12]. 2-(N-morpholino)ethane-sulfonic acid (MES) buffer (Sigma-Aldrich- Merck KGaA, Darmstadt, Germany) 0.1 M was adjusted with sodium hydroxide solution (NaOH 1 M) at pH 4.8–4.9.

### 2.2. Preparation of Mixed Self-Assembled Monolayers 

The sensor slides (SPR Navi-200), comprising a 50 nm gold on glass, were cleaned in a freshly prepared “basic piranha” NH_4_OH/H_2_O_2_/H_2_O solution (1:1:5 v/v) at c.a. 80–90 °C for 10 min. The slides were rinsed with HPLC water, dried with N_2_, and then treated for 10 min in ozone cleaner. For the assembly of the chemical SAM (chem-SAM) on the gold surface, the slides were immediately immersed in a 10 mM thiol solution of 11-MUA: 3-MPA (1:10 molar ratio) in degassed ethanol. The slides were kept in contact with the mixed thiol solution for 18 h at 22 °C under a nitrogen atmosphere in the dark. Afterward, the samples were rinsed with ethanol and mounted in the SPR apparatus, drying the glass back-surface of the chip with N_2_.

### 2.3. Surface Plasmon Resonance Real-Time Functionalization

A BioNavis Multi-parameter Surface plasmon resonance (MP-SPR) Navi^TM^ instrument, in the Kretschmann configuration, was used. The SPR instrument, equipped with two laser sources (670 and 785 nm wavelengths) was used to study the gold surface bio-functionalization in situ, by using the 670 nm source for both sampling areas. A wide angular range (50–78 °C) was measured, with real angular resolution of 0.001 °C. The variation of the plasmon peak angular response was monitored over time. All the experiments were performed at 22 °C in a one-channel cell in which the solutions were manually injected and kept in static conditions. This configuration allows the exposure of a gold area of c.a. 0.42 cm^2^ to be functionalized. The strong gold-sulfur interaction results in the exposure of the carboxylic groups of thiols anchored to the sensor surface. For the immobilization of the biorecognition element, the COOH activation can be performed by the well-known EDC/NHSS chemistry [25,26]. Two sensor slides modified with the chem-SAM were tested by using two different concentrations for the functionalizing antibody. In the first protocol, addressed as protocol A, the modified slide was exposed first to HPLC water to acquire a stable SPR response as baseline. Then, 1 mL of an EDC (200 mM) and NHSS (50 mM) aqueous solution was injected through the cell (internal volume 100 µL, plus 100 µL capillary tubing) and left in contact for 2 h. The surface was subsequently rinsed first with H_2_O and then PBS to inject 500 μL of a 100 μg/mL anti-IgM solution in PBS. The antibody was left in contact until a complete bio-conjugation was achieved, i.e., a plateau response was observed in the sensogram (SPR angle vs time). Then, the sensor slide was rinsed thoroughly with PBS to remove unbound antibodies. This preconcentration step was followed by the injection of 1 M ethanolamine PBS solution (EA) for 45 min. The bio-functionalization was completed with the injection of 0.1 mg/mL bovine serum albumin (BSA) solution in PBS, to gain a more compact SAM, less prone to the nonspecific adsorption [27]. The modified sensor surface was then exposed to an IgM solution in PBS (50 nM) for testing the antibody binding efficacy. Protocol A has been described in detail elsewhere [14]. In the second protocol, addressed as protocol B and adapted from Reference [28] (Figure 2c), the baseline was set by injecting, instead of water, MES buffer through the chem-SAM modified surface. The solution of EDC/NHSS, prepared also in MES, was then injected and left in the SPR cell for 15 min. After the surface was washed with MES, PBS was injected to acquire a new baseline. At this stage, the activated chem-SAM gold surface was exposed to 500 µL of 10 μg/mL anti-IgM solution, and the conjugation was monitored in the sensogram, until a plateau was observed. The succeeding steps involving EA and BSA were performed as in protocol A. Finally, the IgM solution in PBS was injected through the functionalized sensor surface, recording the immunoglobulin binding response.

## 3. Results and Discussion

The amount of anti-IgM capturing proteins immobilized on the gate with the two bio-functionalization protocols A and B was estimated by measuring the SPR shift Δθ occurring when the anti-IgM capturing proteins are conjugated to the activated chem-SAM. The final capturing SAM segregated on the SPR slide is schematically depicted in Figure 1. It is well known that SPR technique ensures the direct correlation of plasmon peak shift with the thickness and optical properties of the medium that contacts the metal surface [29]. The activated carboxylic groups of SAM thiols on the modified gold surface can stably bind antibody primary amines. Thus, when antibodies approach the sensor slide, a new layer can be created at the surface and an increase of the detected angle in the sensogram is observed [30]. The efficacy of each step in the functionalization process can be verified by the SPR real-time monitoring for both the investigated protocols.

The experimental trend for the immobilization of anti-IgM loaded at different concentrations is shown in Figure 2. In protocol A, shown in Figure 2a,b, a saturating trend for the preconcentration of anti-IgM is observed and completed soon after 60 min from the exposure to the solution. The high provision of ligand to the surface entails the typical fast increase of SPR angle within the first minutes of incubation [31]. Then, a slower rate of binding occurs until all the available sites on the surface are covered and an equilibrium state is reached. The injection of the PBS buffer (crossed arrow) removes excess of anti-IgM not anchored to the chem-SAM. A successful binding can be confirmed, as no significant decrease of the signal is registered after the PBS injection. The receptor conjugation is completed by injecting the EA solution, after which the unreacted carboxylic groups are deactivated and the electrostatically bound antibodies are washed over [32].

The SPR signal after the rinsing of EA will be related to the amount of effectively bound receptors as reported in Table 1. This angular shift does not differ significantly from the one recorded in the preconcentration. Moreover, the final step involving BSA does not produce significant changes in the angular response, thus a partial insertion of this blocking agent in the well-packed biolayer can be assumed [12,27].

The sensogram recorded for immobilization protocol B is reported in Figure 2c, with a zoom on the injection of anti-IgM 10 µg/mL in Figure 2d. Independently of the used solvent, the optimization of the procedure can be settled first with a decrease in the reaction time of EDC/NHSS solution, without substantial consequences on the activation efficacy. Indeed, the EDC action is completed within 20 min, so that an extended reaction will only result in a longer functionalization time with no enhancement in the subsequent binding rate of the receptor [33]. Moreover, the control over the pH obtained by means of MES buffer for the activating solution (pH ~5) leads to a more efficient reaction [34]. In a two-step reaction, to gain best results, the first activation step (i.e., EDC/NHSS) should be performed in a MES buffer at pH 5–6, then the pH should be raised to 7.2–7.5 with a phosphate buffer for a more effective reaction with the amine-containing groups of the antibody (anti-IgM) [35]. The pH plays an important role during the immobilization since the ligand is uncharged or positively charged in the preconcentration process to promote an electrostatic attraction between the amino group of the antibody and the negatively charged SAM surface. A pH 0.5–1 units below the isoelectric point of the ligand (typically ~ 8) is needed while preserving the negative charge on the sensor surface keeping the pH above 4 [36,37,38]. Optimal reaction conditions have been chosen in the present work according to the results already reported for other receptors [28].

As soon as the activation solution is washed away by the buffer, the ligand solution in PBS is injected into the SPR cell.

The main observed difference, related to the usage of a more diluted solution, relies on the time required to reach the steady state. Indeed, the first 30 min give a slower increase of the signal (Figure 2d) if compared to protocol A (Figure 2b). In static conditions, the molecule replenishment to the surface will be mostly related to their concentration in the bulk, since there is no flow that opposes the depletion of ligands near the surface [39,40]. Hence, for reaching the equilibrium state at a lower concentration, the time needed for the incubation of anti-IgM is at least 3.5 h. Once a plateau in the antibody binding signal is reached, the buffer is slotted over the cell. After rinsing, the signal does not drop-down thus the binding can be considered effective and the functionalization can be completed through the injection of EA and BSA afterward. A slight increase of the SPR angle is observed after BSA step for the second protocol (c.a. 0.07 °C shift from anti-IgM level). For this different behavior, major adsorption of BSA on the SAM can be assumed considering that the amount of the anti-IgM immobilized is half of that obtained in the first protocol. This means that the anchored receptors are more spaced and BSA can stabilize them by prominent steric hindrance, producing an appreciable signal onto the sensor surface. 

The assessment of sensor surface modification is fundamental when developing a biolayer-based device since the properties of the biorecognition element regarding its orientation, surface density, and activity toward the binding analyte can dramatically influence the assay analytical performances [41,42]. In SPR direct assays, the typical explored analyte concentration returns limit of detections in the 1–10 nM range [43,44], with further improvement only by using nanostructures or sandwich assays [45]. Hence, for testing the response of the functionalized surface at saturating concentrations, in this study, the sensor slides for both protocols were exposed to a standard solution of IgM at nominal concentration of 50 nM (10^−9^ M), recording the corresponding SPR angle variation. The response obtained as angle shift and the equivalent surface density is compared in Table 1. This evaluation is done according to the literature [46], which states the relation between surface coverage (Γ, in ng/cm^2^ ) and the plasmon resonance shift by means of the Feijter’s equation:
$\Gamma =\left(n_{a}-n_{b}\right)d_{a}\cdot {\left(dn/dC\right)}^{-1}$ [47]. Here, the coverage is related to the difference in refractive index between the antibody layer (n_a_) and the bulk solution (n_b_), the average thickness (d_a_) of bounded species, and the refractive index increment (*d*n/*d*C). The difference in average refractive index corresponds to $\left(n_{a}-n_{b}\right)=\Delta \theta \cdot k$, where Δθ is the measured angular shift and k is the wavelength dependent sensitivity coefficient. For thin layer beyond the evanescent field depth (less than 200 nm) at a source wavelength of 670 nm, the equation can be simplified: the product (k·d_a_) is approximated to 1.0 × 10^−7^ cm/deg and *dn/dc* to 0.182 cm^3^/g. Thus, the equation becomes Γ = Δθ·550 (ng/cm^2^) [12,14]. The conversion into a surface coverage expressed in number of molecules per cm^2^ can be performed by considering the molecular weight of the species under investigation. At a fixed analyte concentration, by comparing the total bound IgM obtained with the two protocols, an analogous result is achieved (Table 1). Knowing the average response at this saturating analyte concentration for the same sensing platform used in protocol A [13,14], the reduction to one-tenth of biorecognition elements concentration still produces comparable analytical performances, as can be stated considering the response obtained with protocol B. Although the number of anti-IgM available on the surface is slightly lower (5.9 × 10^11^ particles/cm^2^) with this latter protocol, the available binding sites are indeed enough to measure the same response of protocol A. By fitting the exponential growth relative to the analyte binding, a plateau was obtained correspondent to: Δθ = (0.238 ± 0.005) °C for the exposure in the anti-IgM (100 µg/mL) modified SAM of protocol A, and Δθ = (0.227 ± 0.002) °C for the anti-IgM (10 µg/mL) modified SAM of protocol B. Although the response is comparable in both protocols, one should notice that the 90% of the signal can be reached for protocol A soon after 16 min, meanwhile, 43 min are necessary for protocol B. This highlights the importance of selecting the correct time of analyte exposure within the assay for getting results not affected by time-dependent signals. Interestingly, protocol B leads to an improvement in the functionalization process, drastically reducing the concentration of capturing antibodies in the preconcentration solution, without affecting the assay readout. Moreover, the observed angular shifts for the antibody immobilization is consistent with values already reported in literature [48,49,50]. The main goal in a biosensing platform based on a wide-field approach is that of having a high density of receptor on the sensor surface, for the investigation of extremely low analyte concentrations [2]. These working conditions are hardly tied with low consumption of costly reagents, especially if compared with other miniaturized sensing techniques [42]. As SPR has been largely employed as surface-sensitive technique, many standardized protocols are reported in literature [51,52,53]. The usage of a SAM modified surface allows a controlled antibody binding, and the consequent usage of EA and BSA is well-established [26,54]. However, they were mostly focused on the flow-system facility, which enables the consumption of low volume of reagents while continuous replenishment of molecules to the surface. However, the applicability of these protocols to other sensing methods, such as wide-interface bioelectronics, is not straightforward. The assessment of a robust protocol by means of the SPR apparatus in non-conventional ways is presented here. The authors suggest an improvement in the fabrication of the sensor bio-active surface already tested for an immunoassay application, in the SiMoT device described elsewhere [55,56]. The SPR experimental conditions (i.e., solution volume loaded, the gold area exposed, and the manual injections) have been set to be feasible also for the SiMoT, as well as further bioelectronic platforms. After testing the proposed protocol for the bio-functionalization with a real-time technique like SPR, the modification of the gold electrode used in the electronic sensor can be optimized accordingly.

## 4. Conclusions

In conclusion, a modified bio-functionalization protocol of a gold surface is proposed and compared with the one previously adopted for gold gate electrodes in the ultrasensitive SiMoT biosensing platform. The two protocols are compared using SPR technique. The amount of anti-IgM capturing proteins immobilized on the mm^2^ gate area with the two protocols was estimated by measuring the SPR angle shift Δθ when the anti-IgM capturing proteins are conjugated in real-time to the activated chem-SAM. It was shown that the amount of the capturing antibodies can be reduced at least ten times, up to 10 µg/mL, without affecting the assay analytical performance. The new protocol allows better control of the pH during the different steps of bio-functionalization and reduces the cost of the process, therefore, the cost of the SiMoT platform production. Indeed, this study sets the ground for the assessment of scalable manufacturability of biosensing platform based on EG-TFT devices such as multiplexing single-molecule technologies. 

## Figures and Tables

**Figure 1 sensors-20-03678-f001:**
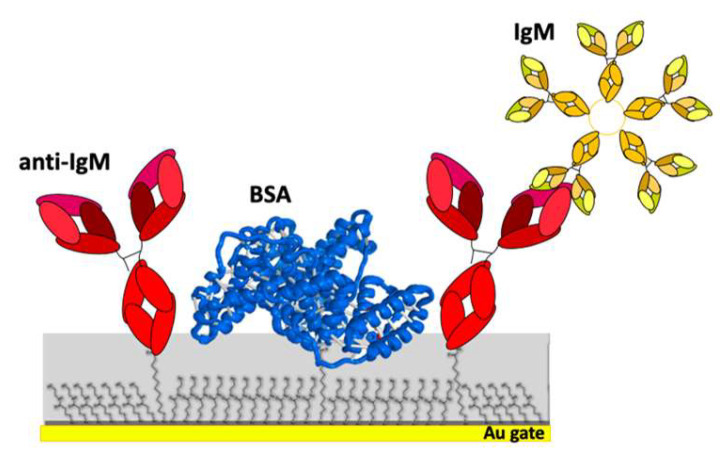
Capturing SAM, comprising both a chem-SAM of activated-and-blocked 3-mercaptopropionic acid (3-MPA) and 11-mercaptoundecanoic acid (11-MUA) and a bio-SAM of capturing antibodies.

**Figure 2 sensors-20-03678-f002:**
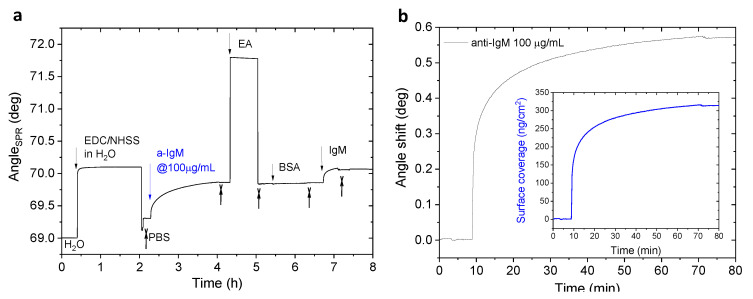
SPR real-time functionalization in which top-down arrows refer to injected solutions and reverse crossed-arrows to phosphate buffered saline (PBS) rinsing steps. (**a**) Sensogram for the immobilization of anti-IgM on the gold surface pre-modified with mixed SAM. (**b**) Zoom of the angular response (angle shift vs time) for anti-IgM exposure at 100 μg/mL and inset showing the corresponding surface coverage (ng/cm^2^). (**c**) Sensogram for anti-IgM immobilization performed with reduced antibody concentration and (**d**) zoom in the 10 μg/mL anti-IgM preconcentration step with surface coverage (ng/cm^2^) in the inset.

**Table 1 sensors-20-03678-t001:** Surface plasmon resonance (SPR) response as angle shift (Δθ) recorded for the anchored anti-human immunoglobulin M (IgM) and IgM exposure in nM range for both protocols. Calculated surface coverage expressed in ng/cm^2^ and number of immobilized molecules for cm^2^ surface area.

		*SPR Δθ* (°)	**SC Γ* (ng/cm^2^)	**SC* (*particles*/cm^2^)
***Protocol A***	*Anti-IgM ^#^* *100* µg/mL	0.53	294	1.2 × 10^12^
	*IgM ^##^*	0.23	127	8.0 × 10^10^
***Protocol B***	*Anti-IgM ^#^* 10 µg/mL	0.27	146	5.9 × 10^11^
	*IgM ^##^*	0.21	116	7.3 × 10^10^

*SC: Surface coverage; # anti IgM MW 150 kDa; ## IgM MW 950 kDa.

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
