# Peer review of "Assessment of Gold Bio-Functionalization for Wide-Interface Biosensing Platforms"

_sensors, 2020, doi:10.3390/s20133678_

Round 1

Reviewer 1 Report

Manuscript details:

Journal: Sensors

Manuscript ID: sensors-833986

Type of manuscript: Letter

Title: Assessment of gold bio-functionalization for wide-interface biosensing

platforms

Authors: Lucia Sarcina, Luisa Torsi, Rosaria Anna Picca, Kyriaki Manoli *,

Eleonora Macchia Submitted to section: Biosensors,

Review comments

This manuscript reports the biofunctionalization protocols for SiMoT biosensing devices, focusing the cost-friendly manufacturability without affecting the sensor performance including the sensitivity and selectivity. The detailed procedures are described well enough about the protocols employed in the work. However, the conclusion needs to be drawn more carefully, particularly about the sensitivity because authors haven’t seemed to try to use the concentration of IgM lower than 50 nM. In general, use of smaller concentration of bio-recognition elements causes lower probability of capturing the analyte molecules, thus producing larger detection limit (poorer sensitivity). Authors need to present why the 50 nM should be used for testing the sensitivity.

Moreover, it was also shown in Figs. 2(a) and (c) that 10 ug/mL concentration of anti-IgM gave the longer time for detection than 100 ug/ml. The detection time for a biosensing device is another issue that should be dealt with as one of the important parameters of performance. In addition, this elongated time of detection may cause another problem with getting time-invariant signals, considering a common problem that a normal SPR machine suffers, i.e., the time variant signals over extended time span. Authors also need to provide the angular resolution of the SPR machine used and the evanescent field depth which may have been used in the surface coverage estimation.       

Author Response

Authors reply #1:

We thank the Reviewer for his/her valuable and constructive comments. In order to address the Reviewer’s comment about the IgM standard solution used to compare the response achieved with protocol A and B, the authors have redrafted the discussion section accordingly (see page 6 lines 230-234). An analyte concentration of 50 nM was chosen based on previous evidences about the SPR sensor performance. At this concentration the signal is significant to compare the two protocols. Herein, the aim of the study presented is to determine which is the minimum biorecognition element concentration requested to achieve a sufficiently high surface coverage (~1011/1012) instead of evaluating the limit of detection of the test. The SPR technique in the direct assay configuration is well-known to have LOD in the 1-10 nM range. Thus, it was out of the scope the exposure to lower analyte concentrations. Moreover, we acknowledge the Reviewer for having pointed out that decreasing the bio-recognition element concentration might cause an increased time for interaction. A comment regarding the issue of time-invariant signal has been implemented (see page 7 lines 253-260): The analyte binding curve was fitted, and a plateau was observed correspondent to: Δθ=(0.238±0.005)deg for the exposure in the antiIgM (100µg/mL) modified SAM of protocol A, and Δθ=(0.227±0.002)deg for the antiIgM (10µg/mL) modified SAM of protocol B. Although the response is comparable in both protocols, one should notice that the 90% of the signal can be reached for protocol A soon after 16 minutes, meanwhile 43 minutes are necessary for protocol B. This highlights the importance of selecting the correct time of analyte exposure within the assay, for getting results not affected by time-dependent signals. Also, a clarification about the SPR resolution and settings used for calculations has been added in the manuscript as suggested by the Reviewer at page 3 lines 123-125 and page 7 lines 237-248.

Reviewer 2 Report

Please, read the comments in the attached PDF file.

Author Response

Authors reply #2:

We thank the Reviewer for his/her valuable and constructive comments. We are glad that the Reviewer appreciated the work. As suggested, a comparison with other approaches used in the SPR literature has been implemented in the discussion. Moreover, we have also stressed the advantages that our protocol bring compared to other biosensing platform applications. We highlighted all changes in the manuscript text file at page 2 lines 67-91 and page 7 lines 267-278.

Reviewer 3 Report

The manuscript by Manoli et al presents optimizations of SPR protocols between two approaches: one classic in-flow conjugation protocol vs a static protocol. Though the manuscript is well written and the steps are thoroughly described with a technical soundness the manuscript lacks of the needed appeal for the readers. 

Major revisions are required to tell a more appealing story for the readers.

Author Response

We thank the Reviewer for the valuable and constructive comment and his/her appreciation for the work. The manuscript has been revised accordingly. We added more details about the novelty and implications of the obtained results, to gain an attractive reading. All the changes are highlighted in the manuscript text file at page 2 lines 67-91

Round 2

Reviewer 1 Report

I believe the manuscript has been revised properly according to the review report.

It seems to be ready for being pubished without any further significant corrections.

Reviewer 3 Report

The manuscript results improved by the addition of the statement in the introduction. I recommend the publication in the present form.